# A Parent-Focused Creative Approach as a Treatment for a High-Functioning Child with Autism Spectrum Disorder (ASD) in Korea: A Case Study

**DOI:** 10.3390/ijerph19137836

**Published:** 2022-06-26

**Authors:** Jung Eun Jeanne Park

**Affiliations:** 1Mogul Institute, Seoul 05649, Korea; jungeup@gmail.com; 2Department of Psychology and Psychotherapy, College of Health & Welfare, Dankook University, Cheonan 31116, Korea

**Keywords:** art therapy, creative arts therapy, autism spectrum disorder, parent-focused, early intervention, parent intervention, psychoeducation, ASD, autism, parent training, parent counseling

## Abstract

This study was conducted on a 6-year-old girl with autism spectrum disorder (ASD) in Korea. The case was initiated in February 2015, and intensive treatment was provided for one year. Then, the case was monitored over the course of 6 years until December 2021. The intervention plan was an art therapy-based treatment plan (Individual Therapeutic Education Plan: ITEP) with two integral foci: (1) creative arts-based parent counseling and education and (2) didactic art therapy with the child. This was a new type of integral approach that was not a standard of care practice in Korea, acknowledging the importance of including parents in therapy and the notion of creative arts therapy. There was no scientific evidence supporting this qualitive approach; however, the intervention was a notable success, sustaining a positive outcome—the intervention (1) reduced the anxiety levels of both the mother and the child in the short term; (2) enhanced the child–parent relationship as well as the home environment of the child while the art therapy-based counseling and education increased the mother’s competence; and (3) enhanced the communicative and adaptive functioning of the child and the mother, with art becoming the supportive breakthrough for their emotional obstacles. The findings suggest that a parent-focused creative approach impacts parental changes and child development: the evidence indicates that parent-driven interventions are a viable option for parents and children with ASD to build a better home environment that supports the child’s development.

## 1. Introduction 

The efficacy of parents’ direct involvement in early interventions for children with autism spectrum disorder (ASD) has been recognized [1,2]. Direct parental involvement encompasses (1) exploring and learning behavioral approaches including floor time and play therapy [3]; and (2) teaching parents the strategies to implement play at home to increase appropriate behavioral responses, and to improve imitation (including gestural imitation) and descriptive language [4].

Focusing on the parent–child relationship in a family’s natural environment to improve the child’s preverbal communication from interaction with their parents [2] is a shared experience that supports the development of the child’s verbal language skills. Social interaction is fundamental for improving language development [5], and the home is the first learning environment of all growing children. Thus, the involvement of parents in treatment plans for children with ASD is a delicate approach that supports the initial building of a secure environment for a child’s development of joint attention and mutuality for socialization and communication [6] with the advantage of building positive emotional processes with their parents in their natural home environment. 

Art therapy has been proven to be effective as a treatment for children with ASD [7,8,9,10,11,12] as well as for parents with children with disabilities including ASD [13]. The well-being of the parent(s) is a cornerstone for the child’s development and growth. Many studies have shown that parents of children with disabilities have higher levels of anxiety and depression [14,15], including the parents of children with ASD [16,17,18]. 

Delivering psychoeducational materials related to ASD to parents has been a critical issue affecting the parents of children with ASD since information and support are scarce in Korea [19]. Parents are often left untreated and rarely receive the individual support that they need [13]. In Korea, individual treatment plans have never strictly included the parents in the child’s treatment.

Including parents in their child’s treatment by providing adequate information about ASD and guiding them is one way to empower parents [20] and reduce their parental stress [21]. Using art with children at home is an opportunity [22] in which art-making processes with one’s hands and materials provide a sensory experience [7] with togetherness, while working with art reinforces (1) nonverbal communication and interaction, (2) regulation and expression, and (3) abstract thinking and play [9,10].

## 2. Literature Review 

One of the core symptoms of the original definition of ASD was conflicts in affect when in contact with others [23], including deficits in facial expressions and the inability to form affective connections with people. Kanner [23] remarked that unlike typically developing children, children with ASD have trouble using communicative gestures. Delays in pointing have been identified as among the earliest signs of emerging ASD [24]; gestural development has also been reported to be related to language ability, as it links nonverbal and verbal communication [25]. In LeBarton and Iverson’s study [25], reduced pointing was observed in toddlers with ASD, indicating the potential for targeting nonverbal communication (e.g., gestures) in early interventions to reduce the risk of language delay and to target early language development. 

However, researchers have also found that the ability to regulate and control emotions and behaviors is the key to learning and improving communication skills [26], social skills, and behavioral functioning [27,28] in children with ASD. Berkovits et al. [27] studied the relationships between emotion regulation and both social and behavioral functioning in 108 children with ASD aged 4 to 7 years old. The study used parent reports conducted using the Emotion Regulation Checklist (ERC, [29]), Child Behavior Checklist (CBCL, [30]), Social Skills Improvement System (SSIS, [31]), Wechsler Preschool and Primary Scale of Intelligence (WPPSI-III, [32]), Comprehensive Assessment of Spoken Language (CASL, [33]), Social Responsiveness Scale (SRS, [34]), and Autism Diagnostic Observation Schedule (ADOS-2, [35]) as measurements. The results showed a moderate relationship between the emotional regulation of children and their ratings on overall social skills (SSIS; *r*(108) = 0.67, *p* < 0.001) and behavioral functioning (CBCL; *r*(108) = 0.67 = 0.74, *p* < 0.001) from year 1 to year 2 [27]. This suggests that emotional dysregulation is associated with core problems related to communication and behaviors in children in their early years with ASD; thus, recognizing their own emotional states and having strategies for managing emotional lability and negativity can be beneficial for children with ASD to build social and behavioral skills with their neurotypical peers. 

Moreover, the findings in Berkovits et al.’s study [27] supported the affective component of the Theory of Mind (ToM) [36]. Understanding others’ intentions, beliefs, and emotions is important to social information processing; thus, the development of a healthy ToM is critical because it is directly related to the ability to understand complex social situations, interpret social behaviors, and process social cues. In other words, ToM is a prerequisite for the social information process and for the social communication skills that children with ASD often find difficult among their skillset [37]. 

In general, children on the spectrum have limited emotional responsivity connected to social relatedness and communication [38]; they also have difficulty comprehending emotions using nonverbal indicators such as facial expressions [39,40,41]. Richard et al. [41] found that facial expressions are related to the development of facial processing and the ability to access and retain memories. The ability to recognize emotions and understand facial expressions has significance in the development of social interactions in children with ASD [42,43]. 

Therefore, understanding the nonverbal affective signs displayed through facial expressions and knowing one’s own emotional state [28,38,39,40,41,42,43,44,45,46,47,48] become a joint experience between children with ASD and others because emotional and motivational responsiveness are related to development [49].

In relation to recognizing affect and emotions, several studies on art therapy have demonstrated art-based treatment to be an effective approach for learning facial expressions [9,10,41,50]. Moreover, art therapy [12] has been shown to improve and build on social interaction, communication, and behavioral skills. The strengths and benefits of art therapy approaches in treatments with children with ASD should be recognized because spontaneous and playful experiences with art stimulate the development of expression, imagination, and abstract thinking in children with ASD [9,10]. Children use art naturally to grow; in other words, children develop with art [51] because art helps children to build their own “sense of self”. 

The notion of parental involvement in therapy in early intervention with children with ASD is not new [3]. However, there are a few studies on creative arts-based parent-focused approaches. Thayer’s [11] study on a Developmental Individualized Relationship (DIR)-based creative arts therapy program with parental involvement suggested statistically significant positive changes in the development of social emotional skills in children. Moreover, Allgood [52] examined the impact on parent’s perceptions of their children with ASD using a family-based creative art therapies approach. Group music therapy helped parents to gain new insights about their children: the approach guided parents to (1) understand the importance of their relationship with their children; (2) identify the strengths of their children; and (3) understand their changing role as parents. 

However, there needs to be more research on this subject: creative arts-based parent involvement in therapy for children with ASD. This case study was sanctioned under the official policies and auspices of the Lesley University Institutional Review Board. The ethical decision to involve parents in therapy was explained at the initial intake process. Informed consent forms were signed along with the agreements on the use of art in academic and research purposes. This case was a seed study for the development of the Creative Arts-based Parent Training (CAPT) program for parents with children with ASD in Korea [21]. 

## 3. Case Description

This case study is presented to illustrate the details of an individually tailored art therapy-based treatment plan (Individual Therapeutic Education Plan: ITEP) for a child with ASD to meet her individual needs. Part of the ITEP included parental education and counseling. This was a new type of direct parent involvement in therapy for children with ASD in Korea. The case demonstrated the parent-oriented change that resulted in emotional development, social adjustment, adaptation, and communication skills in the child with ASD in Korea. The after-treatment course was longitudinally monitored for 6 years. 

### 3.1. Clinical Profile 

April was a 4-year-old girl who had received an ASD diagnosis. Her mother was her main caregiver during the weekdays due to the location of her father’s job. April was diagnosed as having high-functioning autism (HFA) at three different major local hospitals in Seoul, Korea, using the Autism Diagnostic Observation Schedule—Module 2 (ADOS-2, [53]; K-ADOS, [54]) and the Childhood Autism Rating Scale (CARS, [55]; K-CARS, [56]). She exhibited HFA symptoms, including social communication deficits, an unusual way of using language, restricted emotion and recognition of emotions, limited interests, and stereotypies. She did not have unusual sensory interests or tantrums and aggression. She was able to point and had some reciprocal skills. According to Helen and the clinical psychologist’s report, April demonstrated inappropriate social behaviors (e.g., monologues, unresponsiveness when focused on her interests, some echolalia, and the repetition of words). She was reported as having difficulty in understanding emotional words and having a challenging time interacting, sharing, and socializing with her peers at kindergarten due to her limited communication and social skills. According to her speech/language assessments—the Preschool Receptive Expressive Language Scale (PRES, [57]), Receptive and Expressive Vocabulary Test (REVT, [58]) and the Speech Mechanism Screening Test (SMST, [59])—April had delays in her general language ability: unusual responses, unresponsiveness, and expressions. April was recommended for individual speech therapy by her initial doctors. 

### 3.2. Diagnostic Formulation 

Art therapy was suitable for April. April displayed strengths in many areas, especially in her visual–motor integration skills. Additionally, she was in the average DQ (PEP-r, DQ = 90%) and SQ (PEP-r, SQ = 92). Moreover, her motor skills and other general self-help abilities were average. She also did not show any perception- and cognition-related developmental issues. She presented some level of development in communication—she was verbal and had age-appropriate perceptive language, though she lacked expressive language skills. In addition, she was good at listening and following rules. April liked artmaking; she was familiar with the art materials and did not have any sensory problems related to the materials. Her pressing needs were building her ability to communicate and adjust in social situations through the use of words and emotional responses to others so that she could adapt herself in various social situations. She also needed to learn to understand facial expressions and use facial expressions purposefully for emotional interactions. 

## 4. Materials and Methods 

### 4.1. Art-Based Assessments 

As part of her pre-treatment schedule to establish her therapeutic goals, the Expressive Therapies Continuum (ETC)-based art assessment was administrated to April because material interaction reflects the characteristics of the user and their therapeutic needs [60]. Moreover, the Expressive Therapies Continuum (ETC) attempts to understand the functions and structures of the brain through the process of art expression using art materials [61,62,63]. During the art-based assessment, April was able to choose her own art materials and initiate the process with clay and a house figurine that is usually used in sand tray therapy. April wrapped yellow i-clay around the house figure (Figure 1, right). She was receptive and had limited responses—she was not very communicative but engaged in monologues. She was not interactive with the therapist though she was following along and listening. She seemed to enjoy solo pretend play. She showed no aggression and did not have any problems with transition. Based on the assessments, April’s Individual Therapeutic Education Plan (ITEP) with targeted goals (Appendix A) was prepared.

### 4.2. April’s Individual Therapeutic Education Plan (ITEP)

April’s tailored ITEP had targeted short- and long-term goals (Appendix A). The ITEP also recommended music therapy for April to help her vocalization and to build her self confidence in making sounds. Psychoeducation and art therapy-based counseling for her mother, Helen, were part of the clinical decision for April’s ITEP (Figure 2). Intense 1 h individual parent education and counseling continued after April’s 1 h individual art therapy session. After the first 12 weeks, Helen only had short check-ins and received parent counseling on an as-needed basis for the entirety of April’s treatment period, including during the monitoring years. Some of the later counseling sessions during this monitoring period were telecommunication-based. 

#### 4.2.1. Parental Education and Counseling Plan

Helen was previously diagnosed as having Generalized Anxiety Disorder at a local hospital and was in the partial remission stage. She reported a high level of tension, anxiety, and negative emotions regarding herself and her child. Helen was overly cautious. She was the main caregiver for April. Helen partly blamed herself for April’s diagnosis even though she knew it was not her fault. Helen had occasional sleep problems. She was afraid of being the sole figure making decisions for April most of the time due to Brian’s absence during weekdays. She felt overwhelmed by her responsibility over April. Helen received 1 h art therapy-based counseling with psychoeducation for the first 3 months (12 sessions) after April’s art therapy sessions (Appendix B). 

#### 4.2.2. Art therapy with April

Studio art therapy influenced the open art setting for April. The studio model provided various kinds of art materials with a range of fluidity and rigidness, and April chose her own art materials for each session. Sensory-focused art materials, feathers, sand, finger puppets, playdough, i-clay, plaster strips, sponges, paints, finger paints, rollers, makers, pompoms, clay, cookie cutters, pastels, pens, ink, board games (Haim Shafir, Jenga wood block game), bead arts, pencils, and drawing materials were materials chosen by April throughout the intervention period. 

## 5. Results 

### 5.1. Progress 

#### 5.1.1. Sessions 1–6: Initial Stage 

In the initial stage, April made a self-representation of a yellow ducky. She repeated wrapping of the house with yellow clay like she did in the initial art assessment (Figure 1). She focused on her repetitive sand play with the yellow ducky and finger puppets (Figure 3). She liked pressing the sand roller down to make the sand even; then, she would bury the finger puppet (woodpecker) in the sand. She preferred sand play and lacked self-expression though she showed increased interactive responses with the therapist after successfully building a rapport. April had invited her mother to Session 6 but did not welcome her. Helen’s individual counseling and parent psychoeducation were continued until Session 12. The parent counseling session also included couples’ therapy. The couple was also assigned the task of finding proactive ways to change themselves to encourage a better parent–child relationship with April. In addition, the Pixar animation *Inside Out* was recommended for April to watch. 

#### 5.1.2. Sessions 7–17: Active Stage 1

April started adding other art projects after playing with the sand: (1) mixing paints with clay; (2) making a house for the ducky; (3) making a pompom doll; (3) making a cat with clay and the cookie cutter, and then painting it with colors; (4) drawing a window in the ducky’s house, etc. However, she usually wanted to paint or draw after sand play. She was mostly silent or minimally verbal, but she asked for help if she needed it. She also responded to the therapist regularly. April’s patterned sand play with the finger puppet (woodpecker) and the following art project continued until session 19. April learned emotional expression with faces through a drawing series of Elsa with pastels (Figure 4). 

#### 5.1.3. Sessions 18–31: Active Stage 2

April clearly stared showing a difference in her emotions and expression around Sessions 16, 17, and 18. She preferred drawing rather than sand play. By Session 18, April would respond to the therapist 80–90% of the time. She initiated verbal communication and used words to express herself voluntarily. April started talking about her friends at her kindergarten play group. April started biweekly peer-group art therapy with an HFA boy (Sessions 19, 21, 22, 24, 26, 28, 30, and 32). In the peer-group art therapy, the children played board games and either drew or worked with clay together. April was observant of the other child’s art and behaviors. She liked to draw the other child. She showed sharing moments, yielding when they played board games together (Session 22). She was responsive to him and all situations. In individual sessions, April usually wanted to draw with various materials including ink, pastels, and pencils. Pencils were her preferred drawing medium. She voluntarily started art journaling with “orange cat” at home (Figure 5). She liked drawing with her emotional representation, “orange cat” (Figure 6). She liked bead art and continued bead art at home with her mother. April liked to draw her inner voice or thoughts (Figure 7 and Figure 8). Drawings usually included the orange cat, sometimes with written words or phrases. She would draw an Elsa-like girl and reflect herself in her drawing and talk about her kindergarten as well as her friends and teachers. April became talkative, expressive, and imaginative. 

#### 5.1.4. Sessions 31–37: Termination Stage 

During Session 31, termination was discussed with Helen. In mid-October (between Sessions 25 and 26), April was assessed for her speech and language ability. Her report mentioned that April improved her sharing ability with others and improved her expression, including her feelings and status. The speech therapist mentioned that April still needed some practice to expand her boundaries to accept and collaborate with peers in different social situations and work on her communication; however, the report revealed that she had developed both age-appropriate receptive and pragmatic language abilities. April was notified about her termination in advance, and she wanted to engage with sand play and drawings for Sessions 32 to 35. She made goodbye cards and shared them with the therapist in her last session (Session 37; Figure 9). April did not write but drew a picture of a girl pulling out an orange seed from the ground.

### 5.2. Treatment Outcome 

April’s developmental status, including her intelligence, was examined in June 2016 after her last session in March 2016. The WPPSI-III, VMI (Visual–Motor Integration, [64]), BNT (Boston Naming Test, [65]), and CAT-C (Clinical Assessment of Attention Deficit-Child, [66]) were implemented. Additionally, the CBCL (Korean version, [67]), SMS (Social Maturity Scale, [68]), CARS ([55]; K-CARS, [56]), ABC (Kaufman Assessment Battery for Children, [69]; Korean version, [70]), SCQ (Social Communication Questionnaire, [71]), and SRS [34]) were carried out with Helen. April received an above average score (IQ = 116), and her performance in the intelligence area was slightly higher than it was in the language area. All of her test results, including parent-based reports, said that she was in the average, age-appropriate range, as were her visual–motor skills, Boston Naming Test score, behavioral and social development areas, attention, and adaption. Only the report on her visual attention skills recorded that she was slightly below average, implying she could have attention problems. Noticeably, her K-CARS [56] result said that her functional status had improved, suggesting significant changes in social communication without any observable autism-related symptoms. The diagnosis was performed by clinical psychologists in a major university hospital in Seoul. 

### 5.3. Follow-Up Monitoring 

April’s initial doctors diagnosed her again in March 2017. They said that she was not displaying any symptoms related to autism. April and her parents moved to another district in Seoul when April entered the first grade to attend a private primary school in Seoul. April was attending a regular school program and participating in extracurricular activities. When April entered the third grade, the family moved to Cheon-an, Korea, due to Brian’s job as well as due April’s competitiveness and stress due to her obsession to perform well in all areas at school. Helen worried about April’s stress regarding her academic achievements. According to Helen, April was more relaxed in the school environment and with her friends in Cheon-an. Over the years, April developed her habit of drawing her feelings with pencils. April’s art narratives: art journaling to express her egos and daily feelings became her way of maintaining emotional balance and allowed her to adapt and adjust to the situations of daily life. For example, April would draw her daily activities with friends or her feelings for math homework (Figure 10). Sometimes, she would use speech bubbles in her drawings to describe her wants and needs. Drawing became her way of sharing and expressing her emotions. Due to the COVID-19 pandemic, the last in-person meeting was in 2020. Then, telecommunication-based follow-up took place in December 2021. As of March 2022, April entered the sixth grade. 

## 6. Discussion 

### 6.1. Two-Way Approach: Parent Involvement in Therapies with Children with ASD

Many studies have argued the importance of parental involvement, which is vital for changes and development in children with ASD [1,2,72,73]. For instance, Zaghlawan and Ostrosky [4] provided a modified in-home reciprocal imitation training plan using sets of toys, blocks, balls, and stuffed animals, etc., to provide strategies for parents to learn to integrate imitation skills in play with their children. Children with ASD improved their contingent and gestural imitation skills and some descriptive language skills when both parents learned and implemented the strategies in the home environment. Moreover, a parent-focused intervention emphasizing joint attention and social communication [1] suggested that parents’ learning about initial joint attention skills and functional play to support the development of responsiveness had impacted the development of social communication skills in their toddlers. The social role of imitation is learned through social situations with others [74]; thus, the relationship-based parent–child interactions at home are the foundation for children’s social and communication development. These studies reflect the importance of parents in child development and that home is the base—a natural space for children to learn and grow.

April’s ITEP reflected this idea of teaching and guiding parents to improve the home environment to support her emotional development. The plan for Helen had three parts: (1) being psycho-educated on ASD; (2) learning about and using art and creativity to support April emotionally for child development; and (3) counseling for psychological emotional support as a mother of a child with ASD in Korea. Counseling included couples’ counseling sessions to check on psychological wellness as well as the compatibility of the couple to empower them both as parents. Helen and the family needed to change their family structure to support April as well as Helen’s emotional stability. Brian and Helen were supportive parents who were willing to change the parts of their life, including their family situation, and job circumstances, to create a better home environment for April. They agreed that the home and family environment was fundamental for April’s development. 

Coaching and guiding parents through building the competence within them is a way to optimize the natural resource that each child already has in their environment—the family system. Ingersoll et al. [20] said that what parents provide empowers parents and children at the same time, where parents impact their children while feeling self-efficacy. Empowering Helen with information about ASD made her understand April and her developmental issues. Training and counseling reduced Helen’s stress as well as her anxiety. Lee [13] mentioned that individual support for parents of children with disabilities in Korea very rarely takes place. To see positive changes in children, the involvement of parents in therapy and supporting parents’ emotional needs are necessary. In other words, the treatment plan must be a two-way approach that focuses on the parents as well as on the child in order to create new environmental changes for the child for their development. 

### 6.2. Emotional Development: Connecting Affect and Cognition with Art

Emotional development in children with ASD is emphasized because affect and affective connection are related to core conflicts in people with ASD [23]. Rather than contrasting two systems, exploring the interplay and interaction of the affect and cognition systems may help psychological development and growth in children [51]. Art and the artistic process can be a strategy to integrate affective and cognitive aspects of the brain for general development in all children [51]. Brain functions and the activation of different areas in emotional states differ in drawings that represent different states or moods [61,62]. A recent study by Lusebrink and Hinz [75,76] argued that the relationship between large-scale brain networks (LSBNs) and the functioning of the cognitive and symbolic components of the Expressive Therapies Continuum (ETC) affect the adaptivity and connectivity of the brain. This finding suggests that working with art promotes the rewiring of the brain [61,62,63].

In response to the ETC framework, April’s choice of materials in the first few sessions was sensory (consistently including sand), but her way of using materials was sometimes kinesthetic, for example, wrapping of the house in clay, digging in the sand and burying the woodpecker, and pressing the sand roller hard to make the surface even. At the same time, April was using symbols such as the house, woodpecker, and ducky. April’s art had active patterns with preferences as well as the overuse and underuse of ETC functions [63]. This illustrates that she needed to explore different levels of ETC to enhance the functioning of her brain and connect her affect and cognition [61,62,63].

In each session, April initiated her own art-making process and made patterned art reflecting the “self” for development. She engaged in the distinct use of symbols such as “ducky,” “orange cat,” and “woodpecker” to represent “self,” “growing self,” and “mother” while she played with the sand. Her stories from her drawings had a variety of sources from her kindergarten, playground play, grandma’s house, and farms. This part represents the strength of art therapy for children with ASD [9,10]. Moreover, some of April’s artwork had repetitive forms and were patterned in the process, for instance, sand play and then drawing Elsa. This was her own way of reflecting her memories with emotions. Her artwork represented herself and feelings through her choice of art materials, forms, colors, and mark-making. 

Through self-directed activity with self-chosen art materials, April learned to initiate herself. She was an active agent for her own art process in each session with different art materials, and when she started to draw, she learned to express her emotions, feelings, and thoughts through drawings. Drawing encouraged April to balance her ETC functioning in all ETC components, touching on the perceptive and affect level. In other words, drawing helped her at the cognitive and symbolic levels as well as at the perceptive and affect levels of ETC components, meaning that re-networking and re-balancing were taking place in her brain through the art-making process [75,76]. One of the targeted goals for April was to reinforce cognition-related areas (language, etc.) while improving affective functioning. April also improved related areas such as adaptability and social ability, self-regulation, and behaviors; April learned, connected, and expanded the use of symbols in her drawings for her internal process. 

Symbol use by children serves the purpose of communication—this is one of the characteristics of art [51]. April expressed her feelings, beliefs, and expressions with her symbols. This was an innate process, where the art played a role of survival for April [76]. As her symbols transformed along with her treatment, April changed drastically with them. The therapeutic value of the art process with a variety of materials captures the sense of vitality coming from April’s experience as a human, bringing in and connecting the person (April) to not only another person but to “a person within” (self). April witnessed that art could be an internal and external engagement where art brings with it the concept of vitality in experience of children [77]. Stern stated that vitality in experience was necessary for development in children with ASD [77]. 

It was clear that art therapy helped with building a sense of self in April [78]. Using symbols and learning different emotions, emotional differences, and emotional stages clearly guided April in connecting herself to understanding the cognitive and affect parts of her emotions in her life events. The “angry cat” became her first emotional symbol, where the “self” showed an “angry feeling.” Her drawing connected her affect and memory (cognition) into a story with symbols, not only rewiring April brain’s networking but also opening a new means of communication for April’s verbal expressions with emotions. Art and materials brought about positive changes in adaptivity, language abilities, and her overall functioning and behaviors. The art was therapeutic and educational process for April [79]. This was a clear example that her “three systems—making, perceiving, and feeling” [51] (p. 37) were working for her development. The wholeness of the art experience was a meaningful act of sharing and sharpening one’s experience [80] with social opportunities [81]. 

### 6.3. Art and Creativity: Natural Behavior for Human Development 

Creativity “is not a characteristic of a chosen few, but a process that is within all of us” [82] (p. 113). Individuals are a phenomenological byproduct of life experiences in our relational world. Thus, the motivation for development in children arises from life experiences, which are the materials of creative experiences [83], and are related to the regulation of sensory and emotional experiences [84]. 

Furthermore, creative products begin with an individual using personal experience as the basis for approaching a problem [85] (p. 169). Therefore, creativity is related to children’s development and well-being because using creativity supports children’s emotional and physical self, connecting meaning and daily experiences [86]. Here, using art and imagination is a creative solution to form originality within oneself. Reoperating originality in given frames and tools with art can recreate the vital act of balancing its “fit and appropriateness” for life [87] (p. 92) and can connect a child to the self that is alive and creative to find and fulfil one’s sense of self for ongoing development. 

Acting creatively and using creativity are a survival mechanism through making aesthetic choices and experiencing the process of “making special” things [76] (p. 92). This process of creativity and accepting changes helps children with ASD to maintain personal equilibrium in life. This is the natural “healing power of the arts” and creativity [86] (p. 261). April found her own way to express herself through art. Creativity is an innate human ecological behavior to balance one’s inner self in the universe by expressing and representing personal meaning through choices of action. Likewise, Gardner said that children are artists and that they innately use art in their development [51]. In other words, creativity is a purposeful act that is part of the naturalistic practices of humans and serves the purpose for the self and in relation to others in society [76,88] because expressing oneself helps to create feelings of safety and connections with others. Thus, creativity—the courage to try the unknown and face it imaginatively to make changes and form new things—is the key to healing for children. 

## 7. Conclusions

Although the art therapy-based qualitive approach used in April’s case may not add any scientific value, her treatment outcome and status are notable. She improved in all targeted areas, including in communication, socialization, interaction, peer relationships, and behavioral problems when her 1-year-long treatment was finished in March 2016. Her case is still being studied and followed up by researchers, including her initial doctors in Korea. The intervention plan: (1) creative arts-based parent counseling and education; and (2) didactic art therapy with the child represented a new type of integral approach that was not a standard of care practice in Korea, acknowledging the importance of including parents in therapy as well as the focus on creativity and expression in therapy with children with ASD. The use of creativity to focus on affect and emotional development was key for the intervention, which led to positive changes in the child. Moreover, bringing in the competence in parents through psychoeducation and counseling supported the emotional burden of the parents while the counseling connected the family members to build a more profound parent–child relationship. This newly formed parent–child relationship benefited the child and her home environment to promote further changes. The findings suggested that an integral of a parent-focused creative approach could be a viable therapeutic option for children with ASD to build a better home environment that supports the child’s development. 

## Figures and Tables

**Figure 1 ijerph-19-07836-f001:**
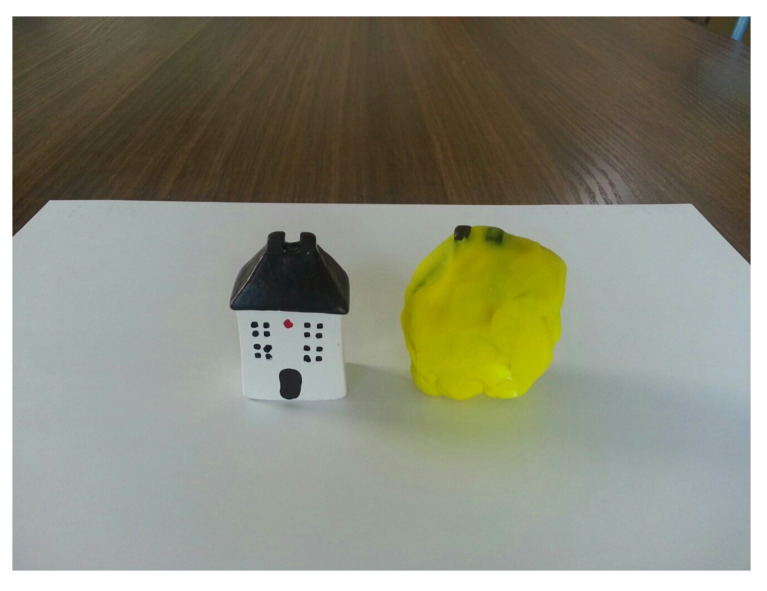
April’s initial art assessment (left: original house figure; right: covered yellow figure).

**Figure 2 ijerph-19-07836-f002:**
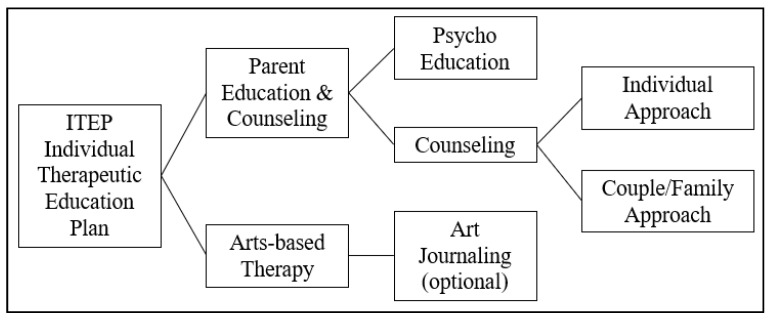
Process of Individual Therapeutic Education Plan (ITEP).

**Figure 3 ijerph-19-07836-f003:**
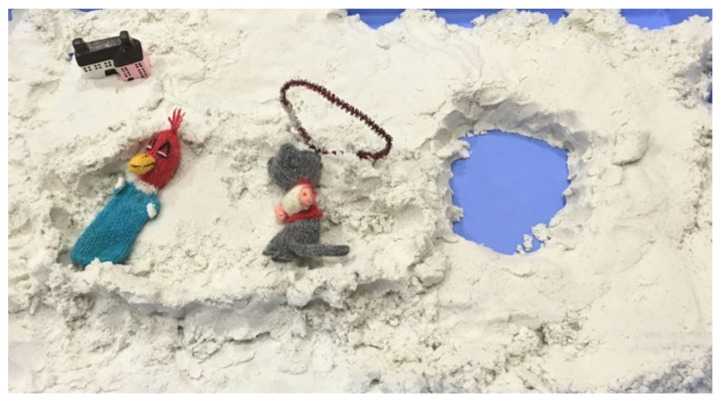
April’s sand play with the finger puppet (woodpecker) and house.

**Figure 4 ijerph-19-07836-f004:**
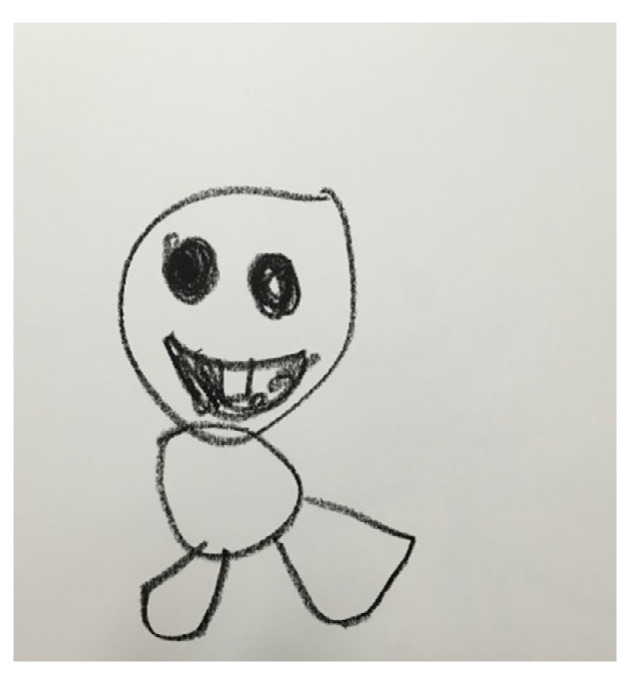
April’s first drawing of emotion.

**Figure 5 ijerph-19-07836-f005:**
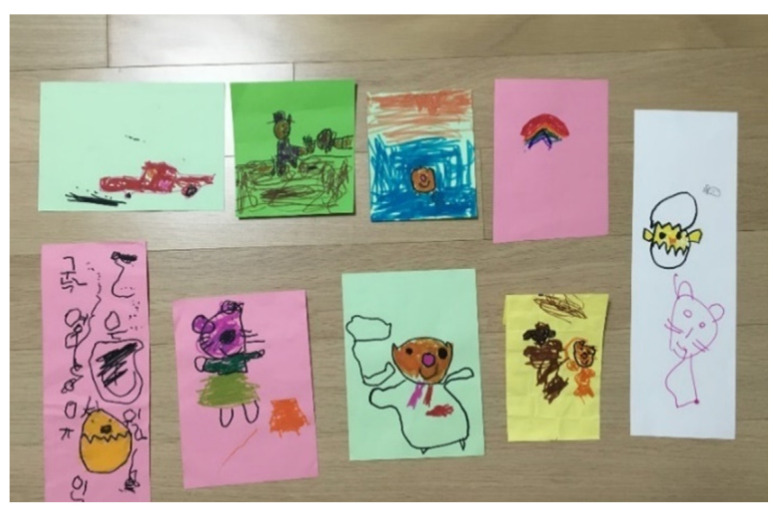
April’s drawing (art journals) with orange cat.

**Figure 6 ijerph-19-07836-f006:**
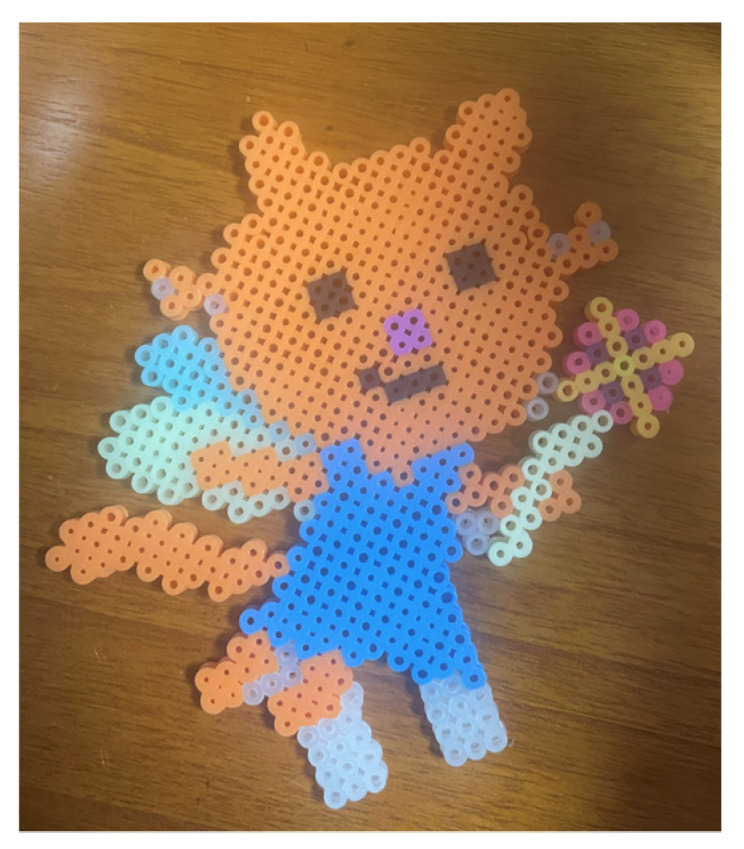
April’s bead art (orange cat).

**Figure 7 ijerph-19-07836-f007:**
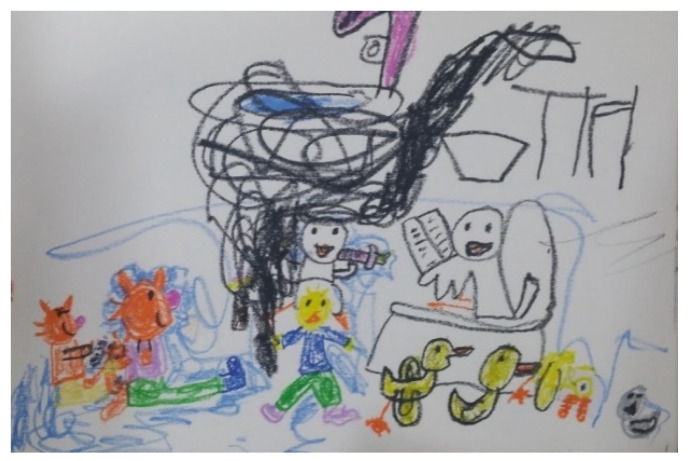
April’s drawings with the orange cat.

**Figure 8 ijerph-19-07836-f008:**
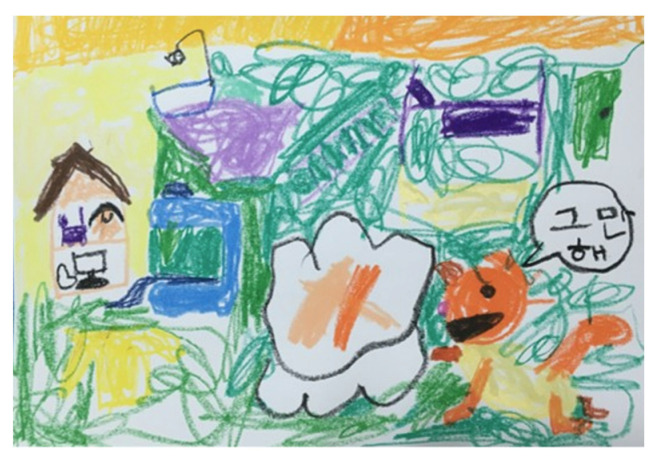
April’s drawings with the orange cat with her inner voice.

**Figure 9 ijerph-19-07836-f009:**
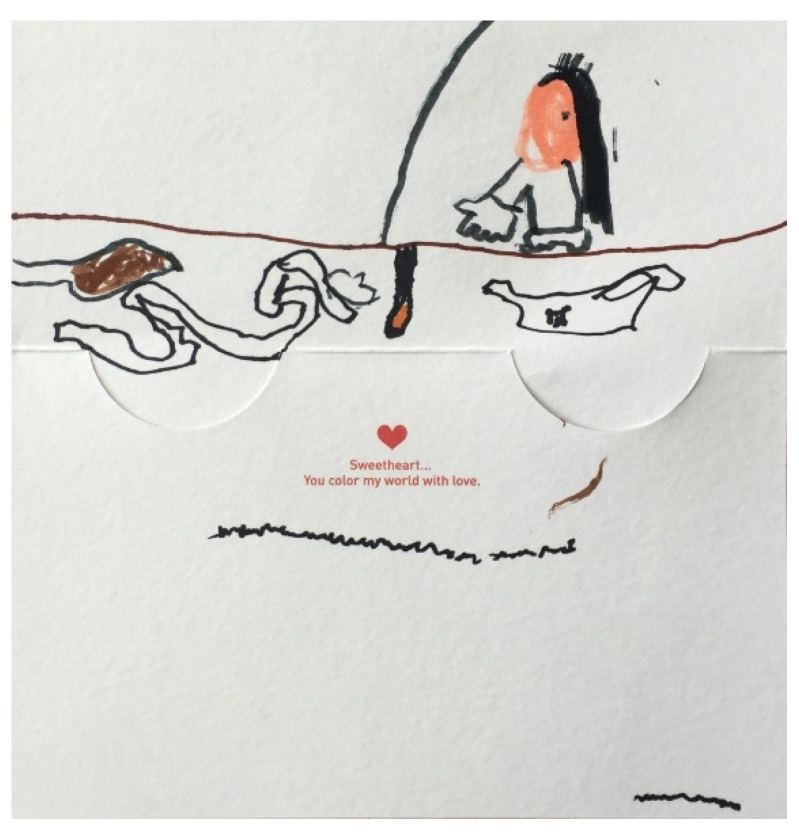
April’s goodbye card to the art therapist.

**Figure 10 ijerph-19-07836-f010:**
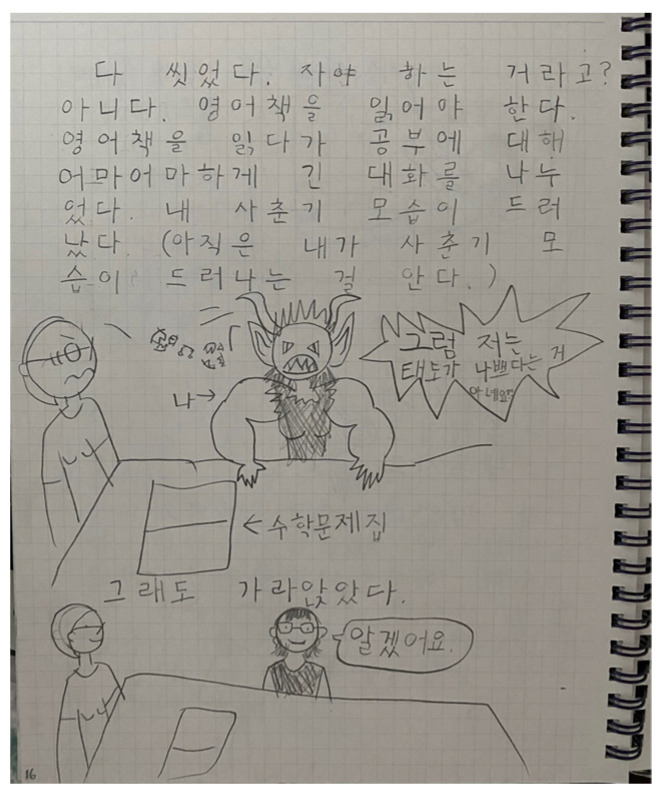
April’s art journaling: cartoon drawing describing April’s feelings about math homework.

## Data Availability

The qualitive data as shown were followed the ethical guideline of human research.

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
