# Peer review of "A Parent-Focused Creative Approach as a Treatment for a High-Functioning Child with Autism Spectrum Disorder (ASD) in Korea: A Case Study"

_ijerph, 2022, doi:10.3390/ijerph19137836_

Round 1
Reviewer 1 Report
Thank you for writing this elaborate case report. I am glad to know that April and her parents achieved favorable outcomes. That being said, I would focus on my role as a reviewer. Here are some of my comments.
1. Case report is too long and contains a lot of irrelevant details that don't add scientific value to the case report.
2. Case report is too unfocused in trying to bring attention to two key points. Based on the review of the manuscript, seems 1. The importance of including parents in therapy and 2. Importance of creative arts therapy. Both of these points are widely accepted when it comes to treating many psychiatric disorders and not Autism Spectrum Disorders alone. As a result, these points don't add real scientific value to current evidence.
3. If the point of this case report is to highlight the importance of practice changes in Korea specifically, then the case report needs to be focused on that point. Adding information about how including parents in the treatment of kids with Autism is not a standard of care practice in Korea might help if indicated.
Author Response
Dear Editor,
First of all, thank you for your time reviewing the paper.
I have checked all your comments and reflected them in revision. I reorganized the structure of the paper, rewrote the parts and shorten the length. Also, reference were edited as well. The highlighted parts were added parts or revised parts for your reference.
Thank you,
Sincerely,
J.

Reviewer 2 Report
I would add part regarding Ethic- the ethical decision to present case, the process of getting ICF from patient and her mother or whole family .This part is missing in the paper. Also I would suggest shortening of the whole paper since it is to long. I really like the style and the content, especially photos. It is very important to have papers like this one.
Author Response
Dear Editor,
First of all, thank you for your time to review the paper.
I have reflected all your comments in the revision.
I have added the parts regarding Ethic- Please check the highlighted parts.
I have shorten the whole paper, and reorganized the structure of the paper.
Again, thank you for your time.
Sincerely,
J.

Reviewer 3 Report
This study is to explore a case study with a the art therapy-based treatment plan for a high functioning child with autism spectrum disorder (ASD). The study motivation and purpose are clear. Overall study processes are valid and proper. Methodology is valid to get a proper results. Followings are some comments to improve the quality of the study.
- In title, A Case Report => A Case Study
- This study mentions a parent-focused creative approach as primary tool to utilize this study. But there is no review in the literature review section. Add a parent-focused creative approach in the literature review section. Or revise it to a proper word such as arts therapy approach.
- Table 1 title is too short. Revise it. It could be a "Process of Individual Therapeutic Education Plan (ITEP)".
- Current table 1's location is not proper. Replace it after line 233.
- Table 1 is not a table, but a figure. Change Table 1 to Figure 1.
- In line 594, delete subtitle of Limitations. It is unnecessary to make such a bold letters with a separate title.
- References should be rearranged.
Author Response
Dear Editor,
First of all, thank you for your time to review the paper.
I have reflected all your comments in the revision.
Yes, - In title, A Case Report => A Case Study
Yes, please check the highlighted parts (in yellow) in the literature review - This study mentions a parent-focused creative approach as primary tool to utilize this study. But there is no review in the literature review section. Add a parent-focused creative approach in the literature review section. Or revise it to a proper word such as arts therapy approach.
Yes, I edited the title- Table 1 title is too short. Revise it. It could be a "Process of Individual Therapeutic Education Plan (ITEP)".
Yes, I moved it- Current table 1's location is not proper. Replace it after line 233.
Yes, I have changed it.- Table 1 is not a table, but a figure. Change Table 1 to Figure 1.
Yes- In line 594, delete subtitle of Limitations. It is unnecessary to make such a bold letters with a separate title.
Yes References should be rearranged.
Again, thank you,
Sincerely,
J

Round 2
Reviewer 1 Report
Thank you for addressing my comments. I would recommend making it more concise and focused so that readers can understand the main points you are trying to convey with this case study.
Line 359: mentions that her testing indicated that she is unlikely to have Autism. As per the current evidence, there is no cure for Autism. This paragraph needs to be rephrased to suggest that her functional status has improved following therapy.
The conclusion section should not cite any references.
Thank you once again for giving me this opportunity to review your work! Good luck!
Author Response
Dear Editor,
Again, thank you for your time reviewing my revision.
I have addressed all your comments and worked with the English service provider from MDPI. Please check the following bullets for your references.
- Yes, I shortened the parts. I made it more concise, especially for the results section.
- Yes, I revised it - Line 359: " said that her functional status had improved, suggesting significant changes in social communication without any observable autism-related symptoms. "
- Yes, I revised the conclusion section, and erased the citations.
Thank you,
Sincerely,
J.
Round 3
Reviewer 1 Report
Thank you for addressing my comments. The author has worked on making the case study concise and focused in a way that clarifies the main message. This manuscript can be accepted in the present form. I wish the author the very best in future endeavors!
This manuscript is a resubmission of an earlier submission. The following is a list of the peer review reports and author responses from that submission.